



# Vertical-axis wind-turbine computations using a 2D hybrid wake actuator-cylinder model

Edgar Martinez-Ojeda[1], Francisco Javier Solorio Ordaz[2], and Mihir Sen[3]

[1,2]Facultad de Ingeniería, Universidad Nacional Autónoma de México, México CDMX, 04510, MÉXICO
[3]Department of Aerospace and Mechanical Engineering, University of Notre Dame, Notre Dame, IN 46556, USA

**Correspondence:** Edgar Martinez-Ojeda (eamo87@protonmail.com)

**Abstract.** The actuator-cylinder model was implemented in OpenFOAM by virtue of source terms in the Navier-Stokes equations. Since the stand-alone actuator cylinder is not able to properly model the wake of a vertical-axis wind turbine, the steady incompressible flow solver SimpleFoam provided by OpenFOAM was used to resolve the entire flow and wakes of the turbines. The source terms are only applied inside a certain region of the computational domain, namely a finite thickness cylinder which represents the flight path of the blades. One of the major advantages of this approach is its implicitness, that is, the velocities inside the hollow cylinder region feed the actuator-cylinder model, this in turn computes the volumetric forces and passes them to the OpenFOAM solver in order to be applied inside the hollow cylinder region. The process is repeated in each iteration of the solver until convergence is achieved. The hybrid RANS actuator-cylinder can be used to model a wind farm since a turbine is now able to take into account the effect of other wakes. The model was compared against numerical and experimental works, wake deficits and power coefficients are used in order to assess the validity of the model. Overall, there is a good agreement of the pattern of the power coefficients according to the positions of the turbines in the array. Wake patterns are also similar at certain distances downwind. The actual accuracy of the power coefficient depends strongly on the solidity of the turbine (actuator-cylinder related) and the inlet boundary turbulence intensity (RANS simulation related); a heuristic approach to correct the results of high-solidity turbines is presented for the case of a wind farm made of small-scale turbines. Thus this method can be used to quickly compute the power coefficients of low-solidity vertical-axis wind farms.

## 1 Introduction

The objective of this work is to develop a computational method for the calculation of a vertical-axis wind turbine (VAWT) farm. This is done by proposing a hybrid numerical method that combines the advantages of a Reynolds-Averaged Navier-Stokes (RANS) solver with an actuator-cylinder (AC) model.

VAWT farms have been gaining popularity recently such as in offshore applications (Hand and Cashman, 2020). Therefore computer models to analyse both wakes and turbine power coefficients are needed. This work will provide the adaptation of the AC model that has been used for VAWTs into an existing OpenFOAM RANS solver. Finally, verifications and validations are included in order to test the model's accuracy. It will be seen that this hybrid method is a potential tool for evaluating wind farms and thus optimize them, that is, in exploring wind farm arrays that otherwise would not have been taken into





consideration. With the help of a wind rose and climate data it is possible to maximize the annual energy yield for a given location and provide insights as to how wind farm arrays behave according to the wind direction, something which has never done before thoroughly.

Interest in offshore applications of VAWTs has grown steadily due to the fact that the percentage of the platform cost relative to the wind turbine itself is dramatically less than that of a horizontal-axis wind turbine (HAWT). Also the omnidirectionality

of the VAWT is highly attractive since there is no need for a yawing system; a HAWT needs a control system to move it with the wind direction, and this reduces its efficiency. VAWTs have benefited from recent improvements in the materials of the blades; in the past they used to have aluminum blades, but those have been replaced by carbon fiber blades which are lighter and more resilient. Improvements in bearing technology have also contributed to their comeback. As for the modeling of the aerodynamics, computing ability has been one of the main problems since most of the numerical methods are computationally

expensive.

Madsen (1982) proposed an AC model which is an analytical solution of the Euler equations coupled by the blade element theory. This model is able to compute the velocity field but it is unable to model the wake recovery. Although it can be extended to model multiple turbines, the non-fading wake renders the model inaccurate. Other simpler models are available in the literature, yet they will not be mentioned here; instead emphasis will be put on actuator models. Full rotor Navier-Stokes are

computationally demanding, therefore making it unfeasible to model a whole wind turbine farm. On the other hand, actuator models rely on modeling the rotor forces through source terms in the Navier-Stokes equations, e.g. applying volumetric forces on the field. This approach proves to be faster since there is no need of a highly refined mesh around the blades of the turbine. Svenning (2010) implemented an explicit actuator disc RANS model to compute the wake of a horizontal-axis wind turbine. The model is stationary. In his implementation; the thrust, torque, inner radius, outer radius and disc thickness are required

as input parameters, hence the term *explicit*. Although simple, its major drawback is that turbines downwind assume constant thrust and torque beforehand, this does not mean all turbines have the same power coefficient though since the pressure drop across the disc is different for each turbine.

A RANS actuator line model (ALM) was recently developed in (Bachant et al., 2018), this model is transient and it can be used to simulate both vertical-axis and horizontal-axis wind turbines. It is able to compute the power coefficient, it is also

provided with a post-processing application that shows the fluctuation of the power as a function of time. Moreover, it includes secondary effects such as dynamic stall, flow curvature, added mass, among others to improve the accuracy of the predictions. Unfortunately, this model cannot be generalized to multiple units. Despite its lack of capacity to model wind farms, it can provide useful information concerning the wake, especially if it is run in the LES mode (Large Eddy Simulation). An actuator disc model for HAWTs can be found in (Nedjari et al., 2020). Full rotor RANS simulations of VAWT arrays can be found in

(Zanforlin and Nishino, 2016; Bremseth and Duraisamy, 2016; Giorgetti et al., 2015; Hansen et al., 2021).

There is another type of ALM model that employs Large Eddy Simulation (LES) instead of RANS. These models are far more accurate when it comes to wake modeling. Shamsoddin and Porté-Agel (2016) have performed a study of a 1 MW VAWT consisting of three blades and a diameter of 50 meters. The study shows results from crosswind and vertical wake profiles as well as turbulence intensity profiles. It is particularly ideal for a verification test of the present work. Finding information about



low-$\sigma$ ($\sigma$ = solidity) wind turbines is rare in the literature; it will be explained later why the AC model performs better for low-$\sigma$ turbines. Abkar (2018) also provides further insight into the wake of a 200 kW VAWT consisting of three blades and a diameter of 26 meters. Whereas the previous mentioned studies were done on one turbine only, LES-ALM results from wind turbine arrays can be found in a recent publication (Hezaveh et al., 2018), interesting conclusions from several array patterns are drawn. Another more complex model is the actuator surface model, which is an extension of the actuator line model. This

approach provides a more realistic flow simulation, a thorough validation of the model can be found in (Massie et al., 2019).

As for experimental results, much of the work is done on small wind turbines either inside a wind tunnel or via towing along a water channel, (although wakes of such small turbines –either wind or water– will not be discussed here) different kinds of works can be found in (Tescione, 2016; Brownstein et al., 2019). Because these turbines have very high $\sigma$, validation is hard to achieve due to the fact that the AC model does not fare well with these kinds of turbines. The only experimental work included

here will be (Araya et al., 2014) which was done on turbine arrays located outdoors. These data come from observations done over a period of months, therefore power coefficients for each of the turbines can be used to validate the present work; unfortunately, these turbines are relatively small and have a fairly high $\sigma$ (above 0.3) due to its large chord-to-radius ratio. Finally, an interesting report from FloWind (Liu et al., 1987) and the U.S. Department of Energy regarding the wakes and power curves from an array of Darrieus turbines is also used to validate the RANS-AC model. These FloWind turbines have

rotor diameters of 17 meters and their $\sigma$ is low (around 0.157), which make them ideal for the validation of the present hybrid RANS-AC model.

## 2   Actuator-cylinder (AC) model for turbines

The VAWT can be modeled as a hollow cylinder upon which radial volume forces $f_n(\theta)$ act. This will create a pressure jump across the entire surface (notice that the cylinder is merely an abstraction, it does not exist materially). The turbine's blades are

responsible for these radial forces. Figure 1 shows a cross section of an infinite long cylinder (in the $z$-direction), the incoming wind velocity is $V_\infty$, $2\epsilon$ is the cylinder's thickness and $R$ is the radius. Madsen (1982) showed that

$$Q_n(\theta) = \lim_{\epsilon \to 0} \int_{R-\epsilon}^{R+\epsilon} f_n(\theta) dr \ . \tag{1}$$

where $Q_n$ is the normal load per unit length exerted on the fluid at angular position $\theta$ averaged over one revolution $2\pi R$. The AC coordinate system is $x$ and $y$. The governing equations are those of continuity and the steady-state Euler equations. Velocities

in the $x$ and $y$ directions are non-dimensionalized by the incoming wind velocity $V_\infty$; lengths are non-dimensionalized by the wind turbine radius $R$ and pressure is non-dimensionalized by $\rho V_\infty^2$.



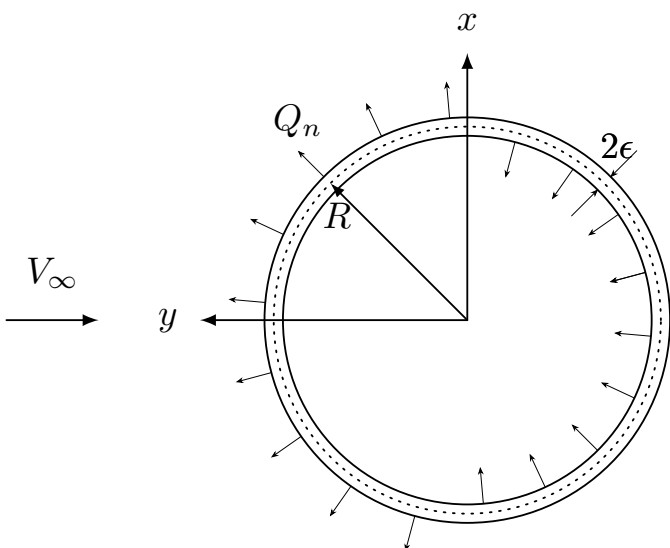

**Figure 1.** Radial forces acting on fluid.

The Euler equations are applied to the entire field and the volume forces are represented by the forces exerted by the blades. The final form of the solution is shown in Eqs. (2a) and (2b), which allow calculation of the perturbation velocities $w_x$ and $w_y$, $N$ is the number of evaluation points, $\theta$ is the angle of the current evaluation point and $\phi$ is a dummy angle used for integration purposes. The turbine rotates counter-clockwise. This form only includes the linear part, although a correction is made to make up for the non-linear terms. When the evaluation takes place inside the hollow cylinder, the term I must be included; on the other hand, if the evaluation point is located in the wake (the leeward part of the cylinder), both I and II must be included in the equation. Li (2017) provides a useful discretization scheme assuming the forces are piecewise constant.

$$w_x = -\frac{1}{2\pi} \sum_{i=0}^{N-1} Q_{n,i} \int_{\theta_i - \frac{1}{2}\Delta\theta}^{\theta_i + \frac{1}{2}\Delta\theta} \frac{-(x + \sin\phi)\sin\phi + (y - \cos\phi)\cos\phi}{(x + \sin\phi)^2 + (y - \cos\phi)^2} \, d\phi$$

$$\underbrace{-Q_n(\arccos y)}_{\text{I}} + \underbrace{Q_n(-\arccos y)}_{\text{II}} \tag{2a}$$

$$w_y = -\frac{1}{2\pi} \sum_{i=0}^{N-1} Q_{n,i} \int_{\theta_i - \frac{1}{2}\Delta\theta}^{\theta_i + \frac{1}{2}\Delta\theta} \frac{-(x + \sin\phi)\cos\phi - (y - \cos\phi)\sin\phi}{(x + \sin\phi)^2 + (y - \cos\phi)^2} \, d\phi \tag{2b}$$

These equations can be put in matrix form using influence coefficients that depend on geometrical variables that have to be computed only once and a column vector $Q_n$. Eqs. (2) can predict the perturbation velocities along the hollow cylinder provided the forces are known, for which the blade element theory is needed.





Figure 2 shows the velocities acting on the aerodynamic center of a blade at position $\theta$. The free stream velocity $V_\infty$ is broken down into its $x$ and $y$ Cartesian components so that they can be projected along the cylinder tangential and normal directions. Note that $\alpha$ is the angle of attack with respect to the chord of the blade, $\omega R$ is the tangential velocity of the blade due to rotation, $V_t$ and $V_n$ are the airflow tangential and normal velocities, $V_{rel}$ is the relative velocity. Sometimes the chord is pitched slightly by an angle $\delta$.

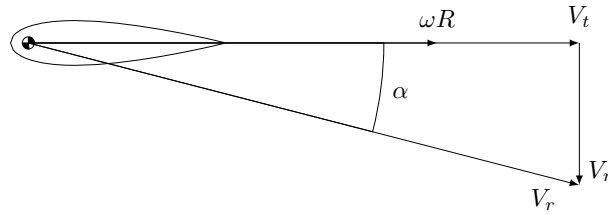

**Figure 2.** Velocity triangle of a blade element.

The normalization of the loads can be found in (Li, 2017) and it will be outlined briefly here. From now on, all variables will be non-dimensional (written in lowercase). The local $x$ and $y$ velocities can be decomposed into the normal and tangential velocities.

$$v_x = 1 + w_x, \tag{3}$$

$$v_y = w_y, \tag{4}$$

$$v_n = v_x \sin\theta - v_y \cos\theta, \tag{5}$$

$$v_t = v_x \cos\theta + v_y \sin\theta + \lambda. \tag{6}$$

The rotational speed of the rotor normalized by $V_\infty$ is $\omega R / V_\infty$, which is a common characteristic parameter of wind turbines called the *tip-speed ratio* denoted by $\lambda$. The normalization of the relative speed and $\alpha$ are

$$v_{rel} = \sqrt{v_t^2 + v_n^2}, \tag{7}$$

$$\alpha = \arctan(v_n / v_t) - \delta. \tag{8}$$

The aerodynamic forces are

$$C_n = C_L \cos\alpha + C_D \sin\alpha, \tag{9}$$

$$C_t = C_L \sin\alpha - C_D \cos\alpha. \tag{10}$$

    The lift and drag coefficients can be obtained from a lookup table as a function of $\alpha$ and Reynolds number. Some common

NACA profiles have plenty of experimental data and can be found in (Sheldahl and Klimas, 1981) at high Reynolds numbers





and for an ample range of $\alpha$. Finally the normal and tangential forces exerted on the cylinder become

$$Q_n(\theta) = \frac{\sigma}{2\pi} v_{rel}^2 (C_n(\theta)\cos\delta - C_t(\theta)\sin\delta), \tag{11}$$

$$Q_t(\theta) = -\frac{\sigma}{2\pi} v_{rel}^2 (C_n(\theta)\sin\delta + C_t(\theta)\cos\delta). \tag{12}$$

The turbine $\sigma$ is given by $\sigma = N_B c/2R$, which can be interpreted as the blades' area per unit length divided by the turbine swept area per unit length. The perturbation velocities can be determined if the forces are known, while the forces also depend on the perturbation velocities. The solution is iterative: first, the perturbation velocities are set to zero, then the aerodynamic coefficients are computed as well as $Q_n$ and $Q_t$. Eqs. (2a) and 2b are used to find the perturbation velocities, and the process is repeated until convergence. To justify the linear correction a factor that depends on the global induction factor and thrust coefficient from the turbine itself is applied to all perturbation velocities each iteration. The correction comes from the momentum theory using empirical equations. The procedure is straightforward and can be found in (Li, 2017; Madsen et al., 2013; Ning, 2016; Cheng et al., 2016).

It is important to keep $\sigma$ low, otherwise the basic assumptions about the model break down since effects such as flow curvature and flow distortion are not taken into account. The model does not guarantee any results whatsoever if high-$\sigma$ rotors are used.

## 2.1 Comparison with Paraschivoiu (2002)

An indirect validation is done using results from a double multiple-stream tube (DMST) model. The DMST model was developed by Paraschivoiu (2002) and it consists of splitting the turbine into multiple stream tubes. Each stream tube has an upwind and a downwind part. Equations of mass, momentum and energy are applied to each tube and an expression for the averaged axial force can be found; equating this axial force to the axial force provided by the blade element theory yields to a local induction factor (relation between the tube's velocity and the tube's inlet velocity) that can be determined iteratively. A more complete description can be found in (Martínez, 2017). A DMST model code has been written based on (Paraschivoiu, 2002).

The DMST validation of three different Darrieus turbines is in Fig. 3. Two of them have diameters of 5 m, and the third has a diameter of 17 m. These results include tip losses. Indirect validation of the AC was done by using straight blades instead of the actual troposkien as shown in Fig. 4. No secondary effects are added. Since the DMST results from straight-bladed turbines have no secondary effects, every vertical slice—the turbine is discretized into several vertical slices or portions—has the same local power coefficient as the others, therefore it is virtually the same as if a two-dimensional DMST were run. The results from the AC and DMST models are compared. The plots show that there is good agreement between the two models. The underlying differences could be because of the experimental corrections used in each model when the momentum theory breaks down, the DMST uses the Glauert correction for induction factors greater than 0.33 whilst the AC uses empirical corrections depending on the value of of the induction factor.



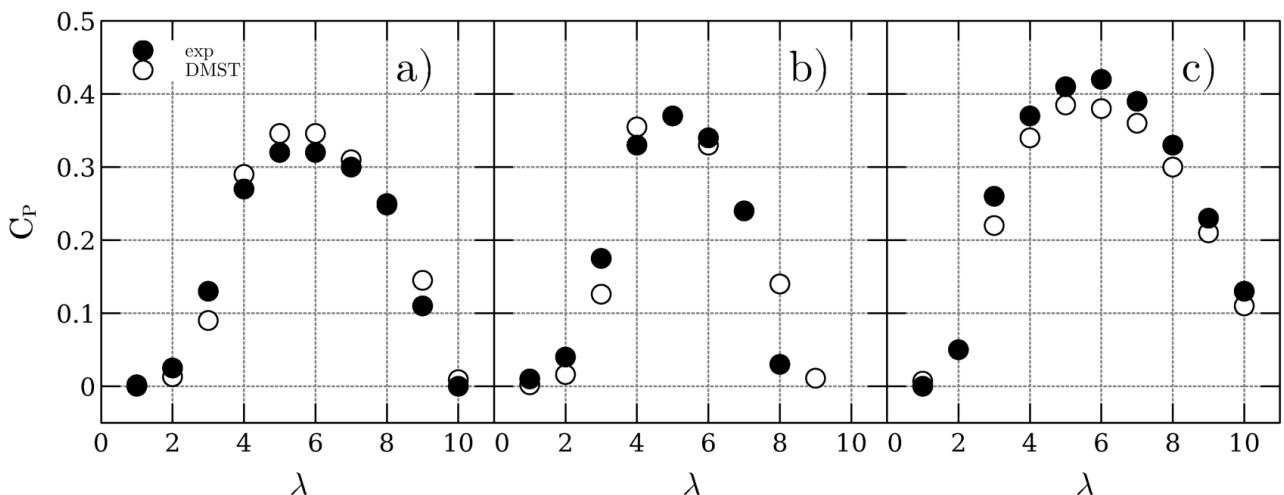

**Figure 3.** SANDIA turbines experimental results: a) Sandia 5 m, 162.5 rpm. 2 NACA0015, b) Sandia 5 m, 150.0 rpm. 3 NACA0015, c) Sandia 17 m, 50.6 rpm, 2 NACA0015.



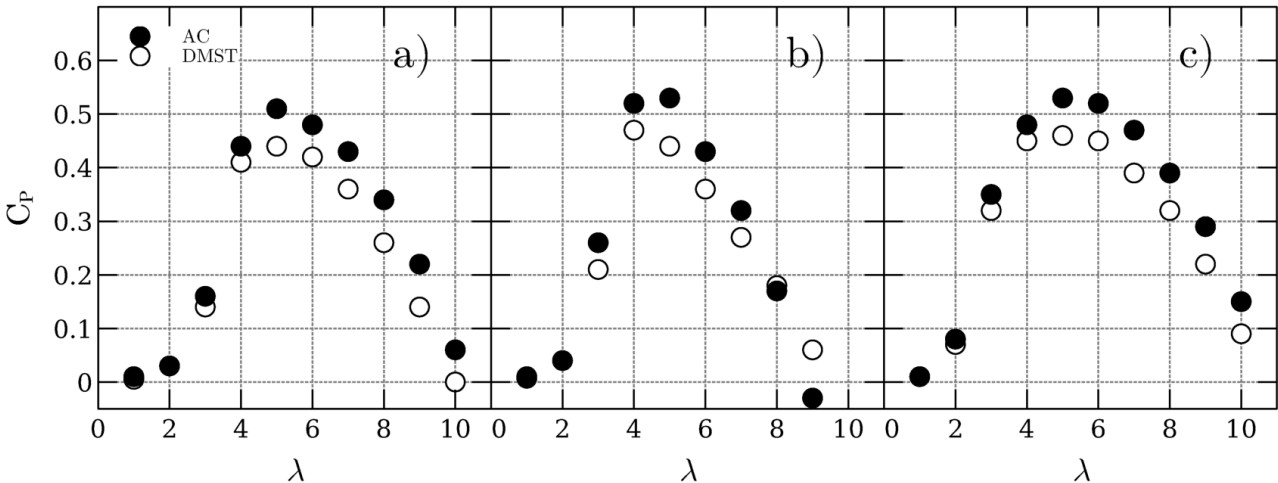

**Figure 4.** Straight-bladed SANDIA turbines. Comparison of models: a) Sandia 5 m, 162.5 rpm. 2 NACA0015, b) Sandia 5 m, 150.0 rpm. 3 NACA0015, c) Sandia 17 m, 50.6 rpm, 2 NACA0015.

## 3    Reynolds-averaged Navier-Stokes (RANS) model for wakes

OpenFOAM is a fluid dynamics open-source toolbox used to solve many different kinds of flows. It provides a plethora of solvers for each kind of flow, e.g. incompressible, compressible, heat transfer, multiphase, combustion, electromagnetics, stress analysis, etcetera. Each solver's settings must be provided with boundary conditions, a mesh, discretization schemes and solution methods for the particular equations. Many solvers can employ either RANS or LES, RANS being the computationally least expensive.

The solver that will be employed is called SimpleFoam (Moukalled et al., 2016) which solves the steady, turbulent and incompressible Navier-Stokes equations with source term options. The source terms will be provided by the stand-alone AC. A $k$-$\epsilon$ turbulence model is used (Launder and Spalding, 1974). This turbulence model is suited for shear flows as well as environmental shear flows; however, it is strongly advised not to use it in flows with large separation and adverse pressure gradients (Bardina et al., 1997; Wilcox, 1998). Svenning (2010) has used the SimpleFoam solver with the $k$-$\epsilon$ turbulence model in his horizontal-axis wind turbine actuator disc model with satisfactory results.





## 4 Proposed hybrid (RANS-AC) computations for turbines with wakes

Given the flexibility provided by OpenFOAM, it is relatively easy to access the velocity field, and it is also possible to create
and manipulate vector fields such as a volumetric force field which will serve as a mean to introduce the forces exerted by the turbine. The hybridization process consists in creating an AC subroutine or program next to an existing OpenFOAM solver and establish communication by passing the velocity field to the AC model, this will compute the loads and thereby the volumetric forces that will be returned to the OpenFOAM's volumetric force field. This operation is repeated each iteration of the solver.

### 4.1 Algorithm

Volumetric forces from Eq. (1) become the source terms in the SimpleFoam solver. Since there is no way of directly getting the force term out of Eq. (1), it is assumed to be constant across the cylinder thickness, so that

$$f_n(\theta) = Q_n(\theta)/\Delta r \tag{13}$$

The normal forces can be projected onto the $x$ and $y$ direction of the computational domain. Using the C++ language, an *actuatorCylinder* class was developed. In programming languages, a class is a kind of data type that contains variables and
functions. This class contains turbine's information such as geometrical parameters, operational parameters, location in the domain and the stand-alone AC model; as well as numerical arrays of velocity vectors and volume forces inside the hollow cylinder. The modification of the existing solver needs minimal modification, namely, create several actuator cylinders in the domain and call the function which adds volumetric forces in the domain.

Figures 5, 6 and 7 show the magnitude, $x$ and $y$ components of both the force field and the velocity field in OpenFOAM.
The units of the volumetric forces are divided by $\rho$ since the solver is incompressible, the Navier-Stokes equations are divided by $\rho$, and thus the source terms will also have their units divided by $\rho$. The flow is from left to right, and the forces are pushing in the opposite direction. The components $x$ and $y$ are also depicted. These forces will eventually slow down the fluid flow, thereby creating a wake downstream. However, both the forces and the flow are time-averaged; the wake will not capture any of the real-life transient effects such as complex vortex shedding and wake meandering.




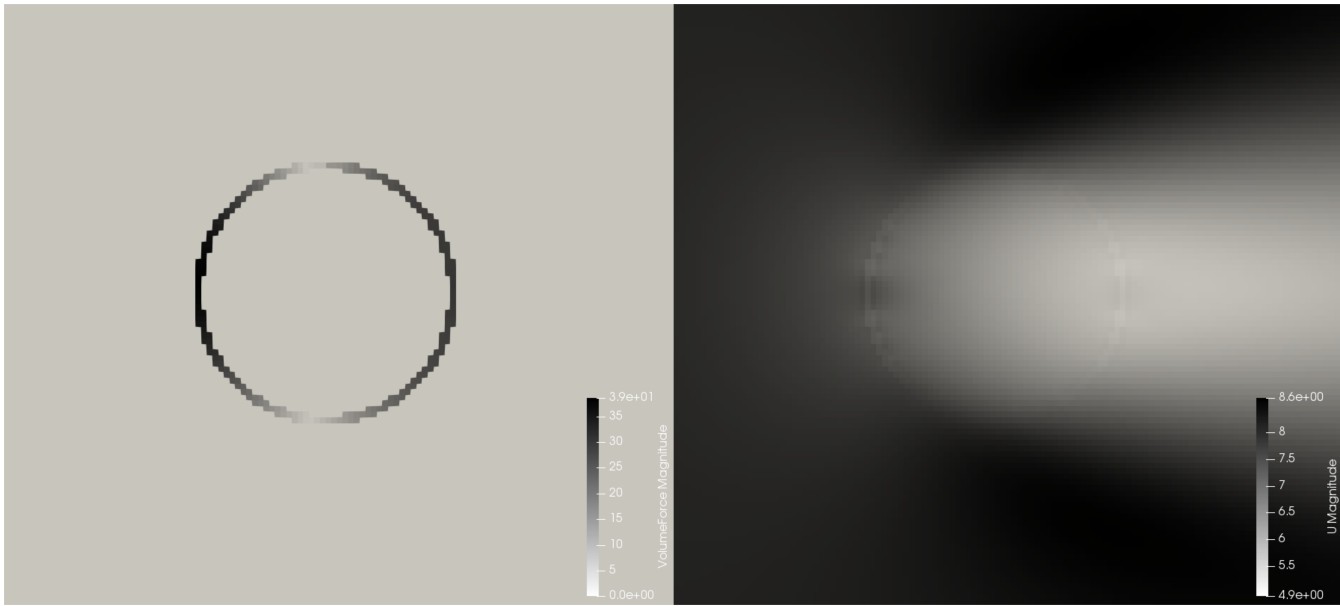

**Figure 5.** Magnitude of force field (left) and magnitude of velocity field (right); volume force is in N m$^{-3}$ divided by density units.

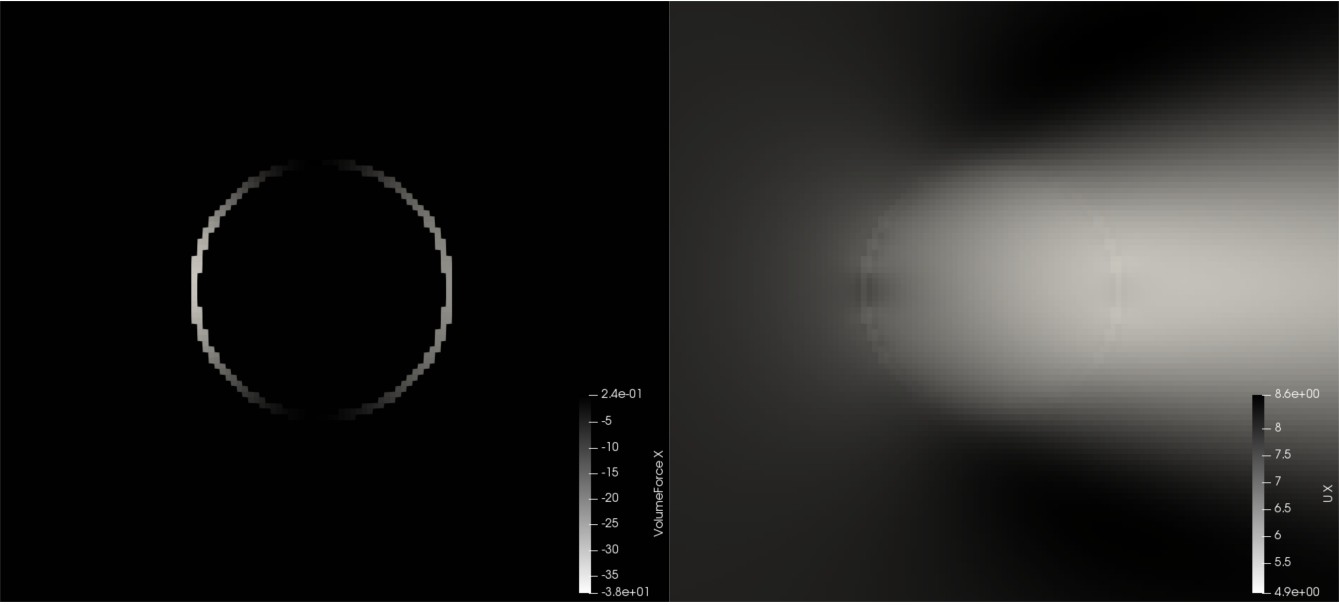

**Figure 6.** $x$-component of force field (left) and of velocity field (right); volume force is in N m$^{-3}$ divided by density units.



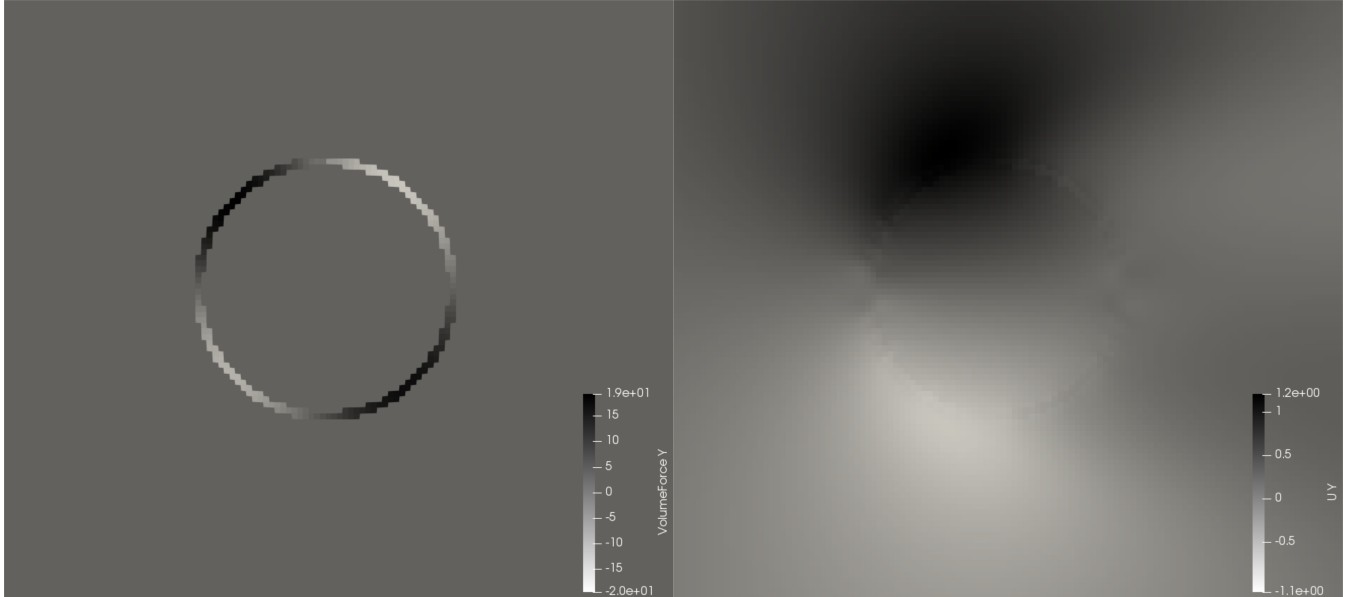

**Figure 7.** $y$-component of force field (left) and of velocity field (right); volume force is in N m$^{-3}$ divided by density units.

## 5   Comparisons with published work

### 5.1   Comparison with numerical simulation Shamsoddin and Porté-Agel (2016)

The study from (Shamsoddin and Porté-Agel, 2016) provides results from an LES simulation using an ALM in the atmospheric boundary layer. Although the simulation is three-dimensional, crosswind profiles of the wake are available at the equator of the turbine. The distance from the ground up to the bottom of the blades is 50 meters. The characteristics of the fictitious turbine are in Table 1.

| $\lambda$ | $V_\infty$ | radius | height | blades | chord | NACA |
|---|---|---|---|---|---|---|
| 4.5 | 9.6 | 25 | 100 | 3 | 1.5 | 0018 |

**Table 1.** 1 MW VAWT characteristics from Shamsoddin and Porté-Agel (2016).

The first part of the study involves power coefficients according to different $\sigma$, this is accomplished by varying the length of the chord. Figure 8 shows the power coefficient curves for four different rotor $\sigma$, namely 0.5, 1.5, 3.0 and 4.5 meter chord. Notice how the prediction is more accurate when the chord length is at its lowest value, this is due to the fact that the blade's aspect ratio is about 200 when the chord is 0.5 m, which is practically an infinitely long blade. A turbine with infinitely long blades behaves the same way as a two-dimensional turbine, the tip losses will have little or no effect on the overall power coefficient.





**Figure 8.** 1 MW turbine power coefficients with varying chord: a) 0.5 m, b) 1.5 m, c) 3.0 m, d) 4.5 m. White dots belong to (Shamsoddin and Porté-Agel, 2016).

The second part of the study includes wake profiles at many different stations downwind using an optimum chord value of 1.5 m. Figure 9 shows how the wake vanishes smoothly; there is an overall good agreement, except in the near wake. The simulation was done with a turbulence intensity $I = 8.3\%$ (the value was taken from the simulation in (Shamsoddin and Porté-Agel, 2016)) and the profiles were taken at the equator's height. The black dots are the results from the RANS-AC; the variable chosen for the wake velocity was simply the ratio of the local speed and the free stream velocity.

**Figure 9.** 1 MW wake profiles at different downstream stations; black dots belong to this work, white dots belong to (Shamsoddin and Porté-Agel, 2016).



Table 2 shows the percentage error of the current work center line wake compared to the LES simulation of the actuator line model. Only at 9D downwind the error becomes 16%; the percentage error becomes relevant in wind farm modeling since most turbines are placed from 6D to 8D apart in case of a checkerboard configuration.

| 1D | 3D | 5D | 7D | 9D |
|------|------|------|------|------|
| 70% | 108% | 62% | 28% | 16% |

**Table 2.** 1 MW center line wake velocity percentage error at various downwind stations.

## 5.2 Comparison with numerical simulation Abkar (2018)

The study from Abkar (2018) is used in order to assess the RANS-AC model. This turbine does exist and it is actually located in Uppsala, Sweden. The study uses an actuator line model with LES. Table 3 shows the characteristics of the turbine.

| $\lambda$ | $V_\infty$ | radius | height | blades | chord | NACA |
|------|------|--------|--------|--------|-------|------|
| 3.8 | 8 | 13 | 24 | 3 | 0.75 | 0018 |

**Table 3.** 200 kW VAWT characteristics in (Abkar, 2018).

Figure 10 shows the wake profiles at various stations located downwind. No data about power coefficients was provided by the authors, neither was the value of the turbulence intensity at the inlet boundary. An $I$ value of 1% was chosen since the study does not mention any atmospheric boundary layer, moreover it is mentioned that the inlet flow is perfectly uniform. Overall, there is a good agreement in the trend of the wakes, except in the near wake. Based on the verification cases presented so far, it appears that the RANS-AC seems to overestimate the wake recovery, probably due to the fact that the model is steady and the wake is almost unidirectional; therefore is unable to replicate the actual stirred motion which probably contains less momentum in the direction of the flow due to the fact that, in actuality, the cross wind component of the wake is significantly stronger.



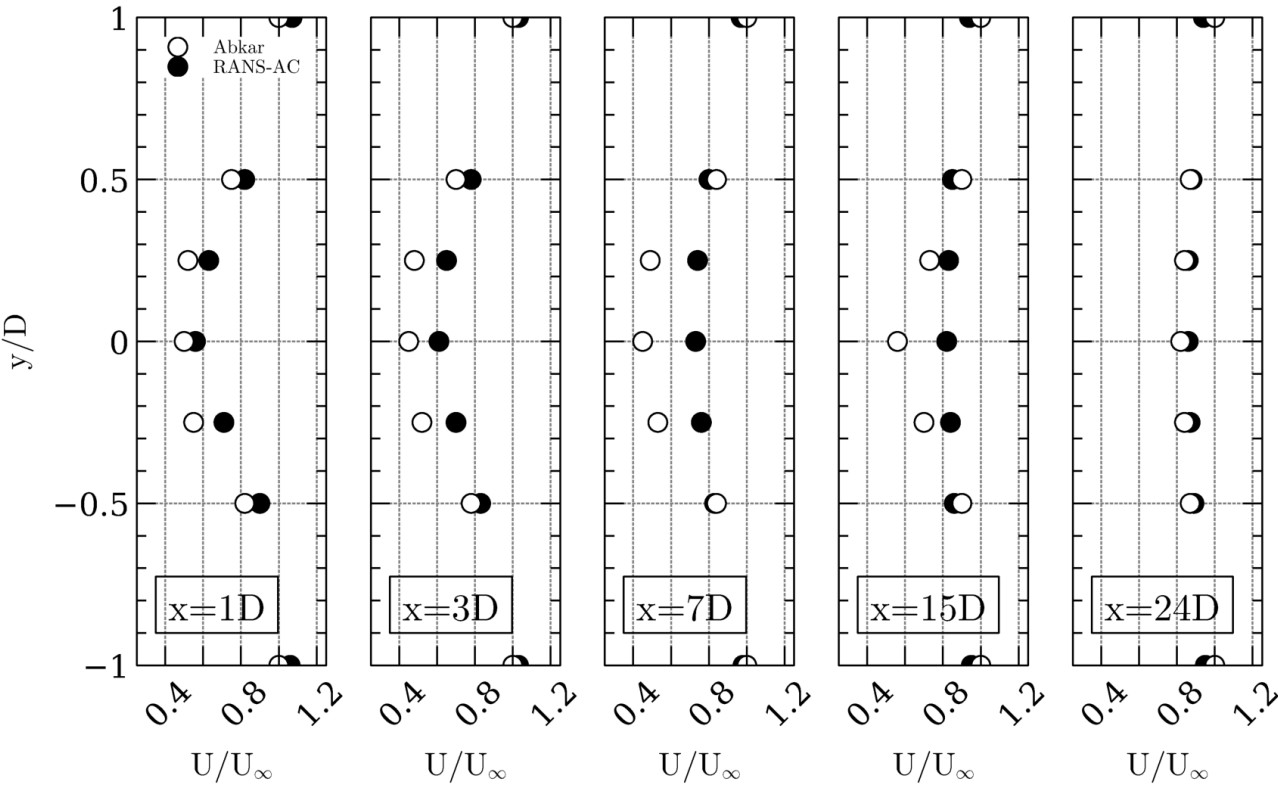

**Figure 10.** 200 kW wake profiles at different downstream stations; black dots belong to this work, white dots belong to Abkar (2018).

Table 4 shows the percentage error of the current work's center line wake velocity compared to the LES simulation of the actuator line model. In this particular case, there was better agreement downwind just behind the wake. The RANS-AC tends to overestimate wake recovery, only at a greater distance downwind both wake profiles match again.

| 1D | 3D | 7D | 15D | 24D |
|------|------|------|------|------|
| 12% | 35% | 62% | 46% | 5% |

**Table 4.** 200 kW VAWT center line wake velocity percentage error at several stations.

### 5.3 Validation with experiment Araya et al. (2014)

Dabiri (Brownstein et al., 2019; Araya et al., 2014) is one of the few to have conducted field experiments using vertical-axis wind turbines. The field experiments conducted at the Antelope Valley in California provide plenty of information and insights as to how vertical-axis wind farms behave with respect to the speed and direction of the wind. Only two cases will be





studied here, namely the case of a four-turbine wind farm and an eighteen-turbine wind farm. The turbines can be mounted and dismounted in order to create new layouts. The turbine used in such field experiment was a three-bladed Windspire turbine; the

diameter is 1.2 m, the chord length is about 0.12 m, the total height of the turbine is 6.1 m and its base is located at 3 m above the ground. According to (Zanforlin and Nishino, 2016), the turbine is kept at an optimal (it is assumed that the turbine is not stalled, and the maximum angle is kept at an optimal value just below stall) $\lambda = 2.3$ below 10.6 m s$^{-1}$ wind speeds. To verify this assumption, the AC model was run at 8 m s$^{-1}$ keeping the optimal $\lambda$. The wing profile was a DU06W200 (Claessens, 2006). The lift and drag coefficients were obtained using the software QBlade (Marten et al., 2013) which is based on the

MIT code XFOIL (Drela, 1989), which relies on the vortex panel method with several add-ons. The local Reynolds number ranges from 100,000 to 220,000 according to the computations of the stand-alone AC model. Figure 11 shows the lift and drag coefficients from a lookup table according to the local Reynolds number experienced by the blades. It is seen that the blade stalls statically at -15° downstroke and 13° upstroke. Theoretically, the blade's $\alpha$ should not go past these limits; it is important to keep in mind that the actual $\alpha$ is not the same as the static angle due to hysteresis effects.

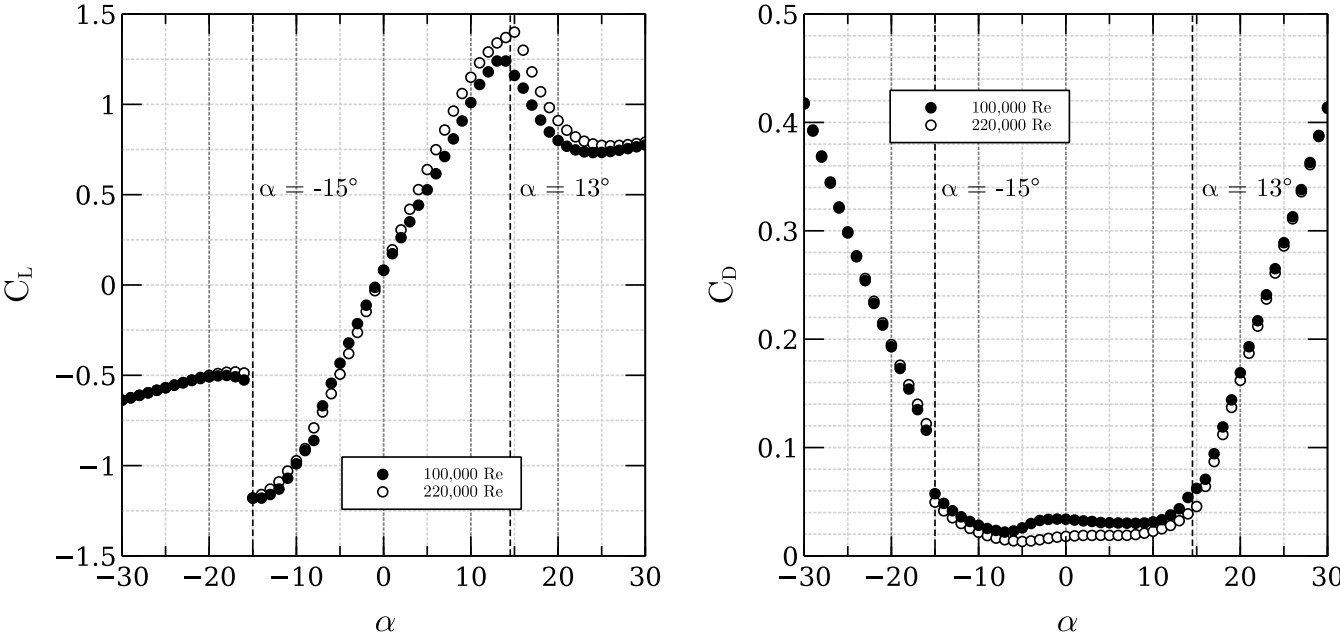

**Figure 11.** DU06W200 lift and drag coefficients from tables; dotted lines represent the angle just before stall.

It was found that the actual $\alpha$ computed by the stand-alone AC model was exceeding the static stall angle according to Figure 11. This is due to the high $\sigma$ of the Windspire turbine and the fact that the model does not fare well with high-$\sigma$ rotors. A heuristic approach to the correction of the $\alpha$ came into mind. This approach consisted in getting a skewed sine function fit to the raw $\alpha$ (the $\alpha$ obtained by the stand-alone AC), then a similar new function was obtained by squeezing the skewed sine function fit just below stall. A corrector function was developed by dividing the adjusted fit by the raw fit, this corrector





function depends of $\theta$ and it is multiplied by the $\alpha$ given by the AC. The angle of attack is then regulated or adjusted so that the blade never stalls. Figure 12 shows how the AC model is able to predict well the $\alpha$ using the heuristic correction. The $C_P$ calculated by the stand-alone AC was 0.14, the CFD simulations show values ranging from 0.14 up to 0.165 for the turbines at the front rows.

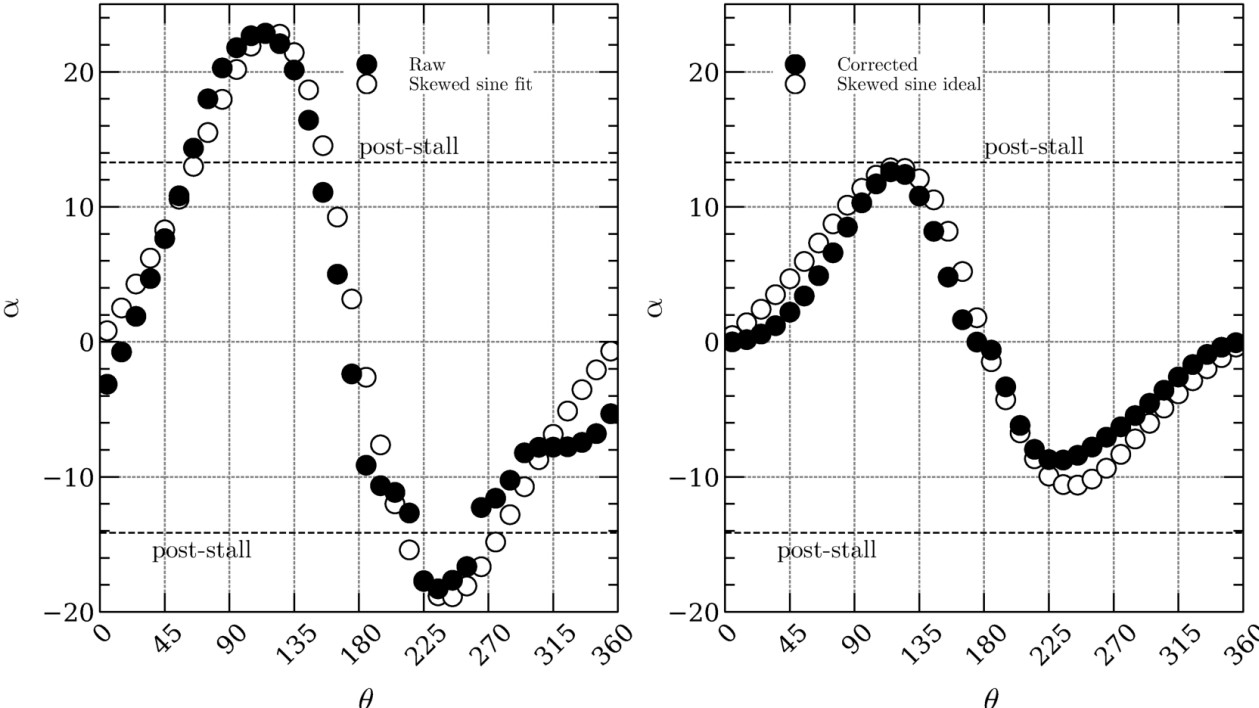

**Figure 12.** Correction of $\alpha$; figure on left shows $\alpha$ computed by AC and its skewed-sinusoidal fit; figure on right shows ideal $\alpha$ and corrected $\alpha$.

The layout of the array configurations is shown in Fig. 13. The first layout belongs to the case of four wind turbines lined up
with the wind coming from the south-west, the latter is the layout of the eighteen turbines, again aligned with the wind coming from the south-west. These turbines are placed in a checkerboard configuration 8D apart, this distance is measured along the $x/D$ and $y/D$ axes. The distance between turbines' axes is 1.65D and the velocity used was 8 m s$^{-1}$.





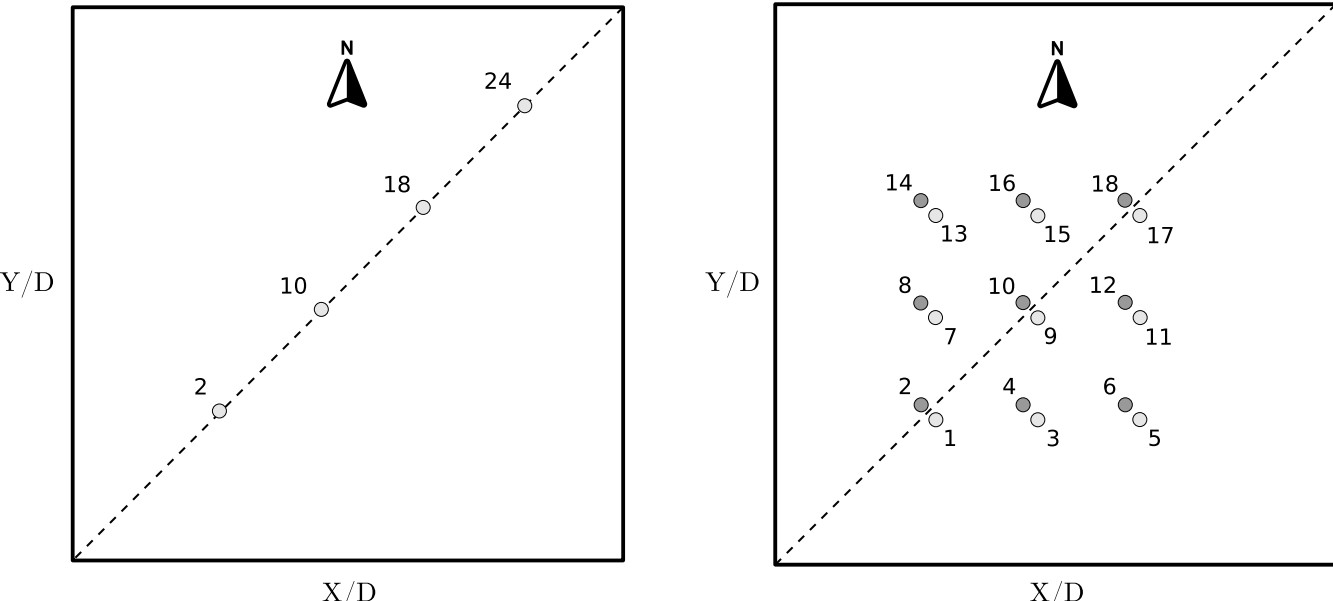

**Figure 13.** Wind farm layouts; wind blows from the south-west at 8 m s$^{-1}$; light colored circles are turbines spinning clockwise, dark colored circles are turbines spinning counter-clockwise.

Next, the power coefficients from each turbine are shown in Figs. 14 and 15. Each figure contains three plots: the actual power coefficient, the normalized power coefficient and the simulation-to-experiment ratio $r_C$. The normalized power coefficient was taken to be the ratio of the actual power coefficient and the power coefficient of the leading turbine; experimental results were normalized with the experimental results, simulations were normalized accordingly too. The simulations were done considering inlet boundary conditions with values of $I = 1\%$ and $I = 5\%$, these values were increased gradually until the results became optimal; a further increase in $I$ turned out to be counterproductive. It is seen that $I = 5\%$ causes the wake to recover faster, thereby achieving higher power coefficients. The actual values of $I$ from the experiments are unknown but a standard deviation up to 10% from the average wind velocity is observed in the csv files provided by data mentioned in (Araya et al., 2014). It is impossible to draw conclusions as to to what extent the $I$ contributes to the validity of the results given the fact that the model is only two-dimensional, increasing the $I$ seems to yield better results but this assumption cannot be generalized, three-dimensional effects also play a big role. There is a great difference in the case of the four-turbine layout, the $I$ seems to affect considerably the power of the turbines. On the other hand, the eighteen-turbine layout does not seem to be affected by $I$, except on the last two turbines. There is an overall agreement in the trend of the power coefficients. The third plot of each figure also contains two continuous lines representing the mean simulation-to-experiment ratio $r_C$, the solid line is the $I = 5\%$ values average, whereas the dotted line belongs to the $I = 1\%$ values average.

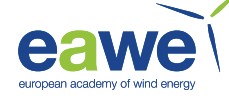


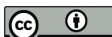

**Figure 14.** Top: four-turbine $C_P$; middle: normalized $C_P$; bottom: simulation-to-experiment $C_P$ ratio.



WIND
ENERGY
SCIENCE
DISCUSSIONS
eawe
european academy of wind energy

**Figure 15.** Top: eighteen-turbine $C_P$; middle: normalized $C_P$; bottom: simulation-to-experiment $C_P$ ratio.

Finally, there is data available from the wake along the center line of the four turbines in Fig. 16. This wake velocity is measured by dividing the speed at each station by the free stream velocity. The data is compared against the results of the simulation, again with $I = 1\%$ and $I = 5\%$. The faster wake recovery can be observed when $I = 5\%$. There is a similar pattern of decay and recovery just before and after the turbine. The vertical dotted lines are the turbines.





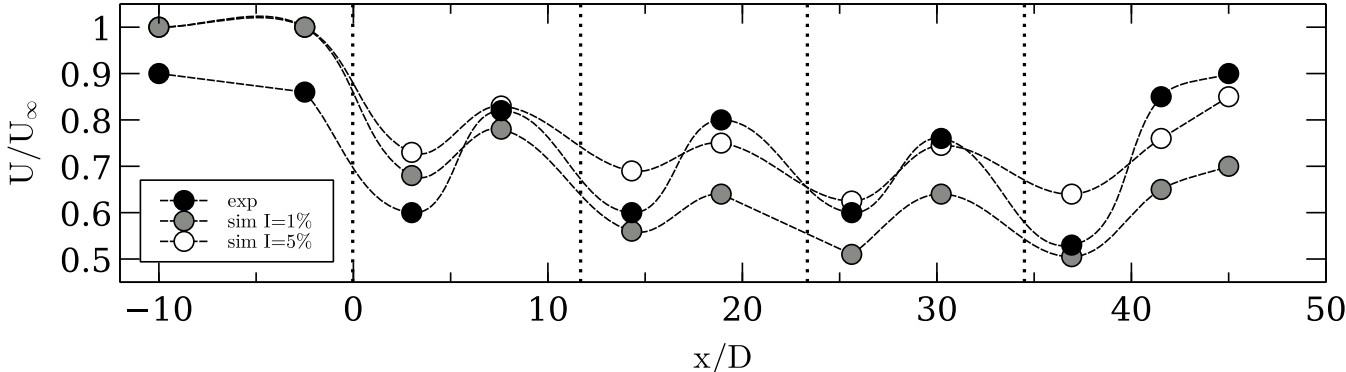

**Figure 16.** Four-turbine center line wake velocity with varying $I$.

### 5.4 Comparison with experiment (Liu et al., 1987)

The next data comes from an experiment done on year 1987, namely on Cameron Ridge in the Tehachapi Mountains. FloWind along with the United States Department of Energy conducted a study on a vertical-axis wind farm (Liu et al., 1987). The study only took nine turbines into consideration, the purpose was to acquire knowledge of turbine wakes and aerodynamic performance of turbine arrays. In the report it is stated that the terrain's aspect is relatively regular. Interestingly, the wind prevails mainly from the northwest (from $135°$ to $315°$) due to thermal effects caused by the heating of the dessert's surface. The farm was built using a spacing of 8D downwind and 3D crosswind, also the rows were slightly staggered in such way that the turbines were all aligned with the most frequent wind direction, which is $128°$ to $308°$ (the report says $308°$, meaning the direction to which the wind blows towards). Figure 17 shows the schematics of the nine turbines used in the study. The figure also contains information about the turbine, which is similar to the SANDIA turbines with troposkian blades. Turbine's ground clearance is 4.6 m.



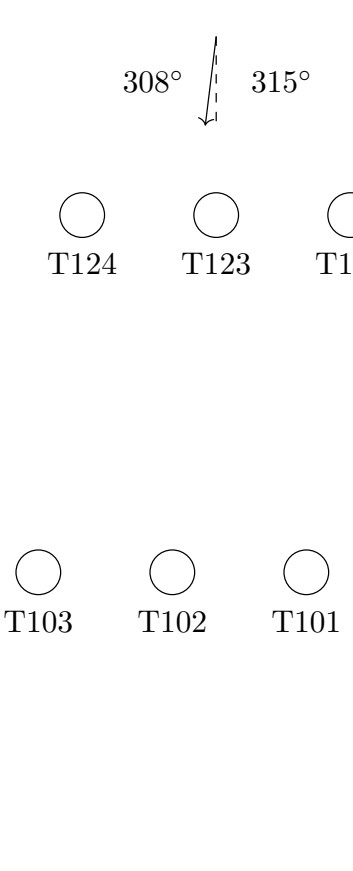

**Figure 17.** FloWind turbine layout in (Liu et al., 1987); T$n$ are Sandia turbines, each 17 m, 2-bladed with section NACA 0015, chord 0.6 m, rpm 53; $315°$ wind direction corresponds to the north-west; turbines are separated by 3 diameters in $x$-direction and 8 diameters in $y$.

An inconvenient part of the study is that not all the turbines are on, meaning that they opted to analyze the power of a turbine under the influence of the rest of the turbines using on and off combinations. This means that the wake deficit includes

a background deficit –the wake deficit when turbines are shut off. Equation (14) shows that the background deficit is subtracted from the actual deficit to yield the true deficit. $S_0$ is the free stream speed and $S_{x/D}$ is the local speed at the current downstream station.

$$V_d = (S_0 - S_{x/D})/S_0 - V_{d0} \tag{14}$$

Information about the background deficit is available in the report for several wind speeds and at different downwind stations

for the case of a single turbine wake. Based on this data, it is possible to compare plots from the section *5.2.2 On-axis velocity deficit* in the report. Since there is no such thing as a background deficit in the numerical simulation (no turbines shut off),



the background deficit $V_{d0}$ was added to Eq. (14) in order to have the experimental data without background deficit. This would allow to make a fair comparison between the simulation and the experimental results. Next, the wake of a single turbine is presented against experimental results considering all of the above details. Figure 18 shows the wake deficit (without
background deficit) behind the turbine. The wake from the simulation proved to be better at distances of 5D and 6.5D and at velocities above 15 m s$^{-1}$. Figure 19 shows the simulation-experiment ratio $r_U$ along with a horizontal dotted line signifying the mean of the results. The value at 11 m s$^{-1}$ in the last plot was removed since it is unusually low, perhaps due to measurement errors. In some cases, satisfactory results can be achieved with a mean deficit close to one. $I = 8\%$ was used in the simulation. This choice was made based on the fact that $I$ can be determined by the ratio of the standard deviation and the mean wind speed
(Gocman and Giebel, 2016); the data from (Araya et al., 2014) showed $I$ above 10% but this value was measured closer to the ground since the Windspire turbine has its center located at 9 m above the ground. More recently, results from an actuator line model (Mendoza and Goude, 2017) using a 12 kW turbine showed that turbulence intensities behind the wake of the turbine could vary from 6% to 10% at a distance 10 to 15 diameters downstream depending on the roughness of the terrain. Since the nine turbines under study in the Cameron Ridge farm are surrounded by other turbines (shut down) $I$ at the inlet boundary
condition will be taken as 8%, which is an average value coming from the results of (Mendoza and Goude, 2017). Although there is a lot of uncertainty about the real value, no information is provided to estimate $I$, the study from (Mendoza and Goude, 2017) uses a similar turbine under similar conditions and it is the best reference available.

**Figure 18.** Single turbine wake deficit comparison with Liu et al. (1987) at various downstream stations.





**Figure 19.** Comparison with Liu et al. (1987) of a single turbine simulation-experiment ratio of wake deficit at various downstream stations; dotted lines are mean values.



Next, the power of turbine T102 with only T123 on is presented. Before continuing, it is reminded that the turbines are aligned along the 308° direction, Figure 20 shows the variation of the power of turbine T012 with respect to several wind

directions. Although the power calculated by the AC is always overestimated, similar trends in the power curves appear. Both curves peak roughly at 17 m s$^{-1}$. Table 5 clarifies the significance of the deviation angle shown in Fig. 20, no deviation angle corresponds to a 308° wind direction. The deviation angle was taken by subtracting the middle value of the wind direction range to 308°. Turbine T012 is slightly shadowed by the wake of T123 in the case 2.5° deviation, the other two cases do not present shadowing from T123, thus yielding similar results in the simulation.

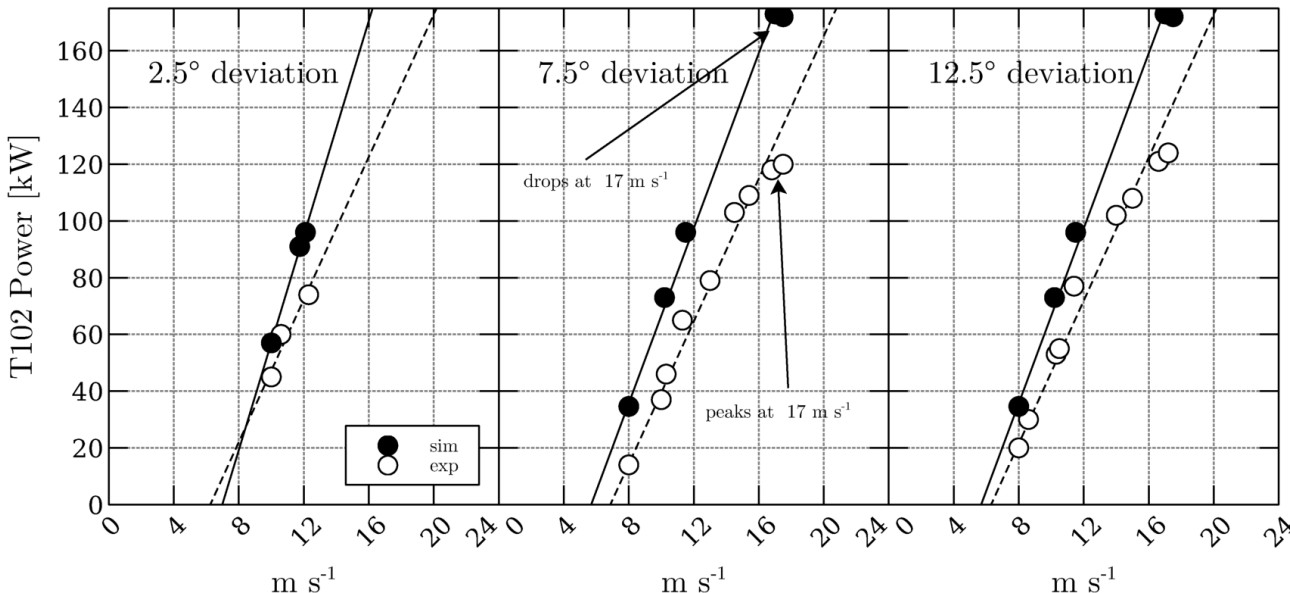

**Figure 20.** Influence of wind direction on T012 power with T123 turned on.

| Wind direction relation to deviation angle | |
| --- | --- |
| **wind direction** | **deviation angle** |
| 308 deg | 0 deg |
| 303-308 deg | 2.5 deg |
| 298-303 deg | 7.5 deg |
| 293-298 deg | 12.5 deg |

**Table 5.** Deviation angle according to wind direction.





The power was obtained by multiplying the power coefficient by $\frac{1}{2}\rho V^3 S$, where $S$ is the rotor's swept area. According to Paraschivoiu (2002), the swept area of a troposkian-shaped rotor is $0.657(4RH)$ . Since the AC is only two-dimensional, the actual area –diameter times AC thickness– cannot be used in the previous formula since it would yield too small a power value; therefore swept area of the real turbine will be used. Power may vary according to the chosen area but the ratio of the power of the turbine under study and the power of an isolated turbine is far more important; in any case, the swept area will cancel out

in the quotient. Table 6 shows the power of the isolated wind turbine from experiments and simulations. Both curves peak at 16 m s$^{-1}$ roughly, the simulation yields 161 kW at 18 m s$^{-1}$.

| Power of a single isolated turbine | | | |
|---|---|---|---|
| **wind speed** | **experimental power** | **simulation power** | **ratio** |
| 8 m s$^{-1}$ | 20 kW | 33 kW | 1.65 |
| 12 m s$^{-1}$ | 77 kW | 106 kW | 1.37 |
| 14 m s$^{-1}$ | 100 kW | 145 kW | 1.45 |
| 16 m s$^{-1}$ | 115 kW | 168 kW | 1.46 |

**Table 6.** Power of a single isolated wind turbine according to wind speed.

Figure 21 shows the ratios of the power values in Fig. 20 and the power of an isolated turbine according to Table 6. It can be seen that in the cases of 2.5° and 12.5° there is good agreement both in magnitude and trend; however, the case of 7.5° deviation shows some discrepancy at free stream velocities lower than 10 m s$^{-1}$. It is important to bear in mind that in the 2.5°

case, turbine T102 is still slightly blocked by turbine T123, therefore it should have the lowest power ratios.



WIND
ENERGY
SCIENCE
DISCUSSIONS

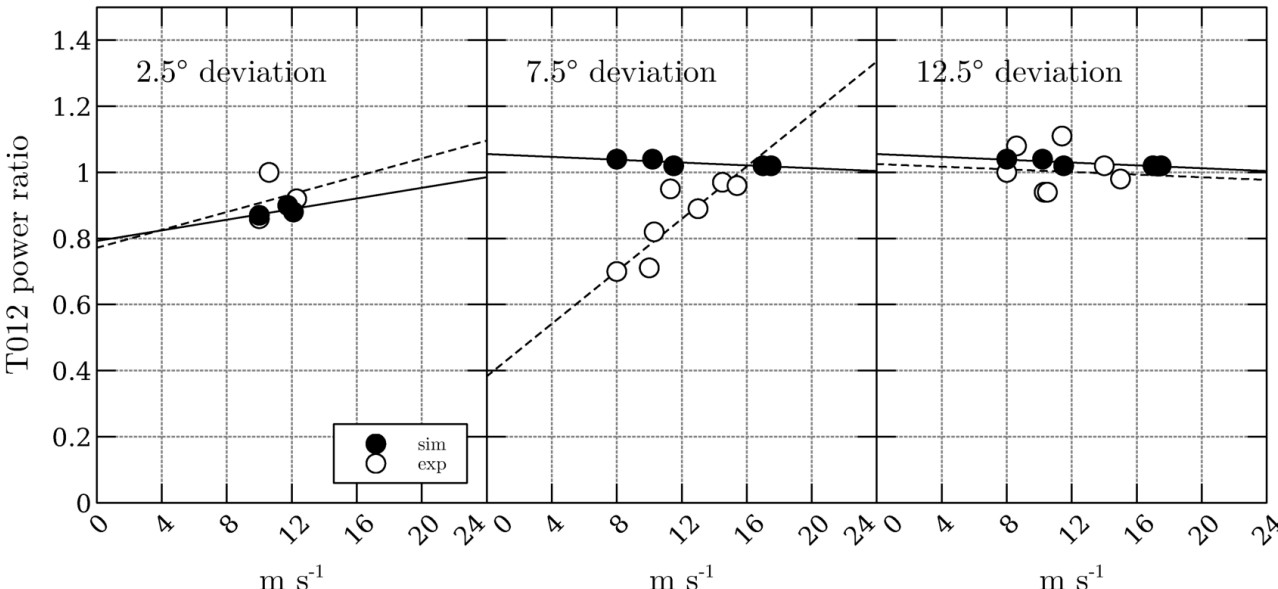

**Figure 21.** Influence of wind direction on T012 power ratio with T123 turned on.

## 6 Conclusions

An implicit-force RANS-AC was integrated in the SimpleFoam solver. The wakes of the model were verified against LES-ALM line models using low-$\sigma$ turbines, a lack of accuracy was observed at in between short and long distances. At distances longer than 8D, better agreement with the LES simulations were found. It is important to notice that the significance of the

wake deficit error becomes more relevant depending on the downstream distance at which the turbines are placed, e.g it is unlikely that a turbine be placed just behind another turbine, therefore the wake deficit error right behind the turbine might not be too important. The validation done against the Antelope Valley site proved to be satisfactory. It was found that the results depend strongly on $I$ at the inlet boundary condition. Higher $I$ values yielded results closer to the experiments; moreover, the trends of the power coefficients were in good agreement with the experiments. The RANS-AC model had a minor tweak in

order to predict the right $\alpha$, this improved the results greatly; this tweak was not necessary in the rest of the present work, only the Windspire turbine in the Antelope Valley needed it. It is therefore advised to take into account the fact that the RANS-AC model is not recommended for high-$\sigma$ turbines, e.g. turbines with high chord-to-radius ratio or a large number of blades. More satisfactory results can be attained when using low-$\sigma$ turbines. Finally, the report from FloWind was used to validate the wake of a single turbine and the effect of turbines on the power coefficient of a particular turbine in the farm. The results from the

wake were satisfactory at distances of 5D and 6.5D downstream, at other distances the mean simulation-to-experiment ratio was 0.6 roughly. The lack of accuracy must be due to the lack of modeling of transient effects such as vortex shedding and





wake meandering, along with three-dimensional terrain and ambient effects. Also the power of the blocked turbine proved to be higher when no blockage was present, just as expected. It is reminded that the present work is not meant to model as accurately as possible the wakes of vertical-axis wind turbines but to predict power coefficients across the entire farm, the current wake
of the model is not said to be superior to other models by any means. Last, the model can work as a powerful tool to research wind farm array configurations in times orders of magnitude less than LES-ALM lines or full-rotor modeling. This is crucial when trying to extract as much energy as possible from a wind farm, thus yielding higher rates of returns. Future work will consider the impact of the wind direction on several wind farm configurations such as checkerboard, staggered checkerboard, even tightly packed configurations employing pairs instead of single turbines.

*Author contributions.* Edgar Martinez-Ojeda was the main author of this work, he developed the stand-alone actuator cylinder code and the actuatorCylinderSimpleFoam solver for OpenFOAM. Mihir Sen contributed to the guidance, suggestions and some of the writing in LaTEX as well as some figures. Francisco Javier Solorio Ordaz contributed to the guidance, suggestions and revision of this work.

*Competing interests.* The authors declare that they have no conflict of interest.

*Acknowledgements.* The main author of this work would like to thank CONACYT (Consejo Nacional de Ciencia y Tecnología) as well as
the OpenFOAM community.





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
