# Peer review of "Vertical-axis wind-turbine computations using a 2D hybrid wake actuator-cylinder model"

_Wind Energy Science, 2021_

## Referee Comment (RC1)

**1 Summary**

The manuscript presents an implementation of the actuator-cylinder model in Open-FOAM. The model is used to study the wakes of vertical-axis wind turbines. The implementation is compared with a number of previous studies, both experimental and numerical. While the topic of the study is relevant for the field, the scientific quality of the paper leaves much to be desired. A major revision, with a fair amount of additional work, is therefore necessary IMO before the manuscript is suitable for publication.

**2 Turbulence**

The main issue with the manuscript is the sometimes cavalier approach to turbulence. Below, I outline my main criticism in this regard.

**2.1 Variation of TI over the computational domain**

The authors give the adopted value of TI without further critical discussion. As turbulence is subject to dissipation, plots should be shown of how turbulence varies over the computational domain (e.g. by plotting TI on a line through the AC, from inlet to outlet).

**2.2 Turbulence intensity is not a free parameter**

In sect. 5.3, turbulence is increased from 1 % to 5 %, 'until the results become optimal'. This is wrong, of course. Apart from the fact that the adopted values are low for the ABL, turbulence is a variable describing a physical state, and cannot be treated as a free parameter. Also, turbulence along a line in the wind direction going through different VAWTs, will vary in a nonmonotonic way. The authors should show how their TI varies along this line. And, again, TI should not be treated as a fit parameter.

**3 Solidity**

While it is understandable that the ACM performs less well a high solidity, the discussion of this issue in the paper is not always clear. On p. 3, sect. 1, the issue is first mentioned. Rather than stating that will be discussed 'later', refer to the appropriate sections. The performance of the ACM as a function of solidity should be discussed more carefully. On p. 3, sect. 1, the high solidity of the VAWT in the paper by Araya et al is mentioned but, nevertheless, there is an extensive comparison with that work. Should the reader conclude that $\sigma = 0.3$ is still acceptable? If so, at which solidities do results cease to be acceptable? The effect of solidity is presented in figure 8, but only chord langths are given. The solidity should be given explicitly, and the results should be discussed more carefully, including the ability of the ACM to generate a result at high TSR and high solidity (lower panels of fig. 8).

**4 Stall**

In sect. 5.3, the discussion of stall and the 'heuristic correction' needs to be improved. The statement is made (p. 16) that the AoA exceeds the static stall angle and this is remedied by squeezing the values of $\alpha$ so that they no longer exceed the static stall angle. I am not convinced that this is justified: at low TSR, the blades *will* experience stall. Why does this physical phenomenon need to be removed from the model? Also, what occurs is dynamic stall rather than static stall. Why then is the focus here on static stall?

**5  Paired VAWTs**

Paired counter-rotating VAWT pairs in close proximity are modelled in sect. 5.3. Were modifications to the model required to handle this case?

**6  Figure 13**

Fig. 13 should have ticks and values for the normalised spatial coordinates, to facilitate the interpretation of the subsequent figures.

---

## Author Comment (AC3)

*Answers to referee no.1*

*We are glad to have your support and time in reading our manuscript and writing your comments. Your remarks have been of great value to the betterment of the manuscript. We hope to fulfill the requirements.*

**2.1 Variation of T.I. over the computational domain.**
**The authors give the adopted value of TI without further critical discussion. As turbulence is subject to dissipation, plots should be shown of how turbulence varies over the computational domain (e.g. by plotting TI on a line through the AC, from inlet to outlet).**
Ans: A plot along the center line will be presented, I is calculated using sqrt((2/3)k) / magU.

Steps taken: This is addressed in line 265, Fig. 11.

**2.2 Turbulence intensity is not a free parameter.**
**In sect. 5.3, turbulence is increased from 1 % to 5 %, 'until the results become optimal'. This is wrong, of course. Apart from the fact that the adopted values are low for the ABL, turbulence is a variable describing a physical state, and cannot be treated as a free parameter. Also, turbulence along a line in the wind direction going through different VAWTs, will vary in a nonmonotonic way. The authors should show how their TI varies along this line. And, again, TI should not be treated as a fit parameter.**
Ans: Data from Araya and Dabiri is used to calculate I. The data contains the average wind speed and standard deviation every minutes from a reference mast. The values of I for different speed bins are presented as well as a Python script which extracts the values of I from the file.

Steps taken: This is addressed in line 215 and Appendix A.

**3. Solidity.**
**While it is understandable that the ACM performs less well a high solidity, the discussion of this issue in the paper is not always clear. On p. 3, sect. 1, the issue is first mentioned. Rather than stating that will be discussed 'later', refer to the appropriate sections. The performance of the ACM as a function of solidity should be discussed more carefully. On p. 3, sect. 1, the high solidity of the VAWT in the paper by Araya et al is mentioned but, nevertheless, there is an extensive comparison with that work. Should the reader conclude that σ = 0.3 is still acceptable? If so, at which solidities do results cease to be acceptable? The effect of solidity is presented in figure 8, but only chord langths are given. The solidity should be given explicitly, and the results should be discussed more carefully, including the ability of the ACM to generate a result at high TSR and high solidity (lower panels of fig. 8).**
Ans: Examples from the literature using AC codes or double multiple-stream tube are listed to explain the highest solidity vales encountered in these models. A validation case against a Windspire turbine is presented, the polars were obtained by using an Xfoil code and details concerning the certainty of these polars are explained according to experimental data. Experimental data was very limited when it comes to Reynolds numbers. The power coefficient curve from the AC follows a very good trend according to the experimental Cp plot, although it overestimates the Cp. Therefore the AC model can still deliver

good trends for high solidity rotors but it is not known whether it predicts the right angles of attack (this causes problems in another validation study later on).

Steps taken: Answer is in line 105.

**4. Stall.**
**In sect. 5.3, the discussion of stall and the 'heuristic correction' needs to be improved. The statement is made (p. 16) that the AoA exceeds the static stall angle and this is remedied by squeezing the values of α so that they no longer exceed the static stall angle. I am not convinced that this is justified: at low TSR, the blades will experience stall. Why does this physical phenomenon need to be removed from the model? Also, what occurs is dynamic stall rather than static stall. Why then is the focus here on static stall?**

The heuristic correction was removed entirely. No dynamic stall model is included because it can cause conflict with the AC model. Although neglecting flow curvature and dynamic stall in the validation case against experimental data from the Windspire turbine still shows a good trend of the power coefficient curve; the fact that the turbine is stalled according to results of the AC model will cause wrong results in the Antelope Valley wind farm of 18 turbines.

It is explained that the unblocked turbines have ironically lower power coefficients than the blocked turbines. Since a blocked turbine sees lower relative velocities and lower angles of attack, these angles will be lower than the unblocked turbines (which are stalled) and thus no stall will be present, thus leading to higher lift forces and thus higher Cps ironically. Data from a rows of turbines in the 18-turbine farm is presented to clear this shortcoming. It is emphasized that this is only due to the fact that the AC model can't predict accurately the angles of attack for this particular turbine

Another case study of a low-solidity turbine is shown: Cps, wake and an 18-turbine farm. It is shown that, since the model predicts well the angles of attack for this low-solidity turbine, the blocked turbines in this case always have lower Cps.

Steps taken: Answer is in line 150 and Section 3.3.

**5. Paired VAWTs.**
**Paired counter-rotating VAWT pairs in close proximity are modelled in sect. 5.3. Were modifications to the model required to handle this case?**

Ans: the heuristic correction was removed.

**6. Figure.**
**Fig. 13 should have ticks and values for the normalised spatial coordinates, to facilitate the interpretation of the subsequent figures.**

Ans: The figure showing the layouts will be corrected.

Steps taken: Fig. 6 and 7 shown the corrected figures.

---

## Author Comment (AC6)

**Answers to referee 2.**

*We are glad to have your support and time in reading our manuscript and writing your comments. Your remarks have been of great value to the betterment of the manuscript. We hope to fulfill the requirements.*

**1. The abstract is too long and general, failing in highlighting the value of the authors' work.**

Ans: This will be corrected and more emphasis on the advantages of this model will be shown.

Steps taken: lines 15 and 50 highlight the importance of this work.

**2. The introduction fails in highlighting the novelty of the authors' work. It is unclear why the proposed method should be preferred to other approaches available in the literature. On top of that, the latter are presented in a confusing way. More references should be added.**

Ans: The advantage is that this is a model with higher fidelity than those of simple potential flows like Rankine bodies, although it takes more time, it is still much less than full rotor RANS simulations where the mesh around the blades has to be resolved. Its implicitness is also an advantage.

**3. From the Reviewer point of view, it would be more effective to merge sections 2 and 3 into section 4, in order to have a more compact overview of the proposed AC model.**

Ans: Indeed, all AC topics are moved into one section: stand-alone AC, linear correction, stand-alone AC validation, RANS-AC implementation and RANS-AC verification against stand-alone AC.

**4. In section 4, where the AC-RANS model is presented, many information is missing. In particular, the algorithms used for the sampling of the inflow velocity from the CFD field and for the insertion of the volume forces into the grid should be clarified.**

Ans: an algorithm presenting the implementation details is added.

Actions taken: the algorithm is in line 170.

**5. No information about the model numerical set-up (mesh, numerical methods, number of iterations, convergence criterion) are provided. Has a mesh sensitivity been performed?**

Ans: mesh, turbulence model, boundary conditions, number of iterations and convergence of the solution and the Cp of the turbine is added too. Two meshes are studied based on a square enclosing the turbine and the thickness of the AC. The coarse mesh yields good results and therefore it will be used throughout the work.

Actions taken: the details are shown in Section 2.5

**6. No information about the AC model formulation is available. How was the polar data used for the simulations obtained? Are any aerodynamic models such as dynamic stall or flow curvature included?**

Ans: The details are shown in Section 2. It is explained why these additional models are not included. Dynamic stall tends to conflict with the AC and flow curvature is computationally expensive because a new virtual profile has to be computed as well as its lift and drag coefficients at new angles of attack, this would imply calling a vortex panel method each azimuthal position times the number of iterations.

Actions taken: the polars are in Section 2.3. Line 150 explains the lack of D.S. and flow curvature.

**7. In the Reviewer's view, it would be nice to split the results in two different section, one for the single-turbine cases and one for the multi-turbine ones**

Ans: This is done by using Sections 2 and 3.

**8. Line 52: How come the ALM cannot be used for multi-turbine simulations?**

Ans: We didn't mean to say that ALM cannot be used for multi-turbine simulations, the model of Bachant is not scalable to multiple-turbines.

**9. Line 61: Is there a quantitative definition for low-solidity?**

Ans: Yes, it is included in Section 2.1, above line 105.

**10. Figures 3 and 4 should be splitted in 3 sub-figures to improve readibility**

Ans: These figures will be excluded. Multi-column or multi-row figures will include a), b), c) and so on.

**11. Section 5.3: Was flow curvature considered somehow? In the Reviewer experience, neglecting such effect for relevant chord-to-radius ratios such as the one considered (c/R=0.2) might introduce a notable error in the results, especially if the static stall angle values were used for tuning as in this case.**

Ans: Go to number 6. The AC overpredicts the power coefficient compared to experimental data of the Windspire turbine but follows a good trend. However it seems to overpredict the angles of attack according to the polars. This will cause issues in later sections and a thorough explanation is given.

**12. Figure 13: it would be nice to add an arrow with the wind direction**

Ans: The figure was redrawn and it is now Figs. 6 and 7.

**13. Figure 14: the legend is not easy to locate, it is suggested to add a frame to it**

Ans: This figure was redesigned and frames will be added to legends when possible.

**14. Line 272: typo, "dessert" is written instead of "desert"**

Ans: Thank you for the correction.

---

## Author Response (AR1)

**Response to Alessandro Bianchini**

Thank you very much for taking your time to read this.

We inform you that we have made the major changes according to the referees' requests. Each of them have been informed already.

Changes made were the folllowing:
a) The abstract was shortened.
b) The aim of this work was made more noticeable in the introduction.
c) The whole implementation and validation of the actuator cylinder was explained thoroughly.
d) The reasons why dynamic stall and flow curvature are included.
e) Problems with high-solidity turbines are cleared throughout the manuscript.
f) Details about the meshing procedure, mesh sensitivity, boundary conditions were explained.
g) Some figures were drawn again with the requirements of the referees.
h) Appendices regarding the procedure used to obtain turbulent intensity values from real data were included.
i) A link to a video showing the results in Paraview is also included.

We have made our best make the paper as clear as possible to the readers.

*Edgar Martinez-Ojeda, Francisco Javier Solorio Ordaz, Mihir Sen.*

---

## Author Response (AR2)

**Response to Alessandro Bianchini**

Thank you very much for taking your time to read this.

We inform you that we have made the minor changes:

An explanation of how the chord-to-radius ratio (which affects the solidity) of the turbine affects the solidity was added. It is explained why the flow curvature alters the boundary layers and both the lift and drag coefficient of the airfoil. Evidence from a reference that we included shows that increasing the chord-to-radius ratio of the turbine is detrimental for the power generation.

We already include valid values of solidity for different models such as the double multiple-stream tube model and the actuator cylinder model.

We believe that we explain clearly how the values of solidity affect the power coefficient of the turbine. Unfortunately the Windspire turbine is somehow difficult to analyze with any of the simple models mentioned above. We aim our research to large wind turbines (offshore), as we have observed that the actuator cylinder model yields much better results when the solidity is low.

We appreciate all of your feedback which was invaluable for the completion of the manuscript.

*Edgar Martinez-Ojeda, Francisco Javier Solorio Ordaz, Mihir Sen.*